# Densely Connected Neural Networks for Nonlinear Regression

**DOI:** 10.3390/e24070876

**Published:** 2022-06-25

**Authors:** Chao Jiang, Canchen Jiang, Dongwei Chen, Fei Hu

**Affiliations:** 1Department of Civil Engineering, Monash University, Clayton, VIC 3800, Australia; chaojiang0728@gmail.com; 2Department of Data Science and Artificial Intelligence, Monash University, Clayton, VIC 3800, Australia; canchen.jiang@monash.edu; 3School of Mathematical and Statistical Sciences, Clemson University, Clemson, SC 29641, USA; 4Institute of Atmospheric Physics, Chinese Academy of Sciences, Beijing 100029, China; hufei@mail.iap.ac.cn; 5College of Earth and Planetary Sciences, University of Chinese Academy of Sciences, Beijing 100029, China

**Keywords:** neural networks, DenseNet, concatenation shortcuts, feature reuse, nonlinear regression, relative humidity prediction

## Abstract

Densely connected convolutional networks (DenseNet) behave well in image processing. However, for regression tasks, convolutional DenseNet may lose essential information from independent input features. To tackle this issue, we propose a novel DenseNet regression model where convolution and pooling layers are replaced by fully connected layers and the original concatenation shortcuts are maintained to reuse the feature. To investigate the effects of depth and input dimensions of the proposed model, careful validations are performed by extensive numerical simulation. The results give an optimal depth (19) and recommend a limited input dimension (under 200). Furthermore, compared with the baseline models, including support vector regression, decision tree regression, and residual regression, our proposed model with the optimal depth performs best. Ultimately, DenseNet regression is applied to predict relative humidity, and the outcome shows a high correlation with observations, which indicates that our model could advance environmental data science.

## 1. Introduction

With the increasing trend in environmental dataset size and complexity, data science has become popular in environmental applications [1]. In environmental data analysis, regression is a useful technique in prediction. Many studies focus on forecasting environmental parameters to address environmental issues, incorporating air pollution, climate change, and global warming. For example, ref. [2] predicts nitrogen dioxide concentrations with a land-use regression method to obtain the spatial distribution of traffic-related air pollution in Rome. Ref. [3] forecasts the river water temperature by regression to further analyze the possibility of future projections considering climate change. Furthermore, ref. [4] proposes a regression approach for greenhouse gases estimation.

Regression analysis statistically models the relationship between the dependent variable and the independent variable. Linear model is the most common form of regression analysis. It is used to model linear relationships and includes General Linear Regression [5], Stepwise Linear Regression [6], and linear regression with penalties, such as Ridge Regression [7], LASSO Regression [8], and Elastic Net Regression [9]. However, nonlinear relationships are more common and complicated in the real world. Therefore, nonlinear regression analysis gains a lot of attention [10,11,12]. There exist many tools to model nonlinear relationships, such as Polynomial Regression [13], Support Vector Regression (SVR) [14], and Decision Tree Regression (DTR) [15], where SVR and DTR are popular nonlinear regression techniques. Nevertheless, SVR often takes a long time to be trained on large datasets, and DTR is extremely non-robust and NP-complete to learn an optimal decision tree [16]. From the late 1980s, people began to use artificial neural networks (ANNs) for nonlinear regression since a neural network with a single hidden layer can approximate any continuous function with compact support for arbitrary accuracy when the width goes to infinity [17,18,19]. According to the universal approximation theorem, the regression accuracy of ANNs heavily depends on the width of the single hidden layer. However, the impact of depth on the accuracy of neural networks are not considered in this classical theorem [20].

Different from ANNs with a single hidden layer, deep neural networks (DNNs) trend in increasing the number of depth (layers) of neural networks, aiming at significantly improving the accuracy of models. In the last decade, there emerge many works on regression tasks using the deep learning-based method. Specifically, ref. [21] proposed a nonlinear regression approach based on DNNs to mimic the function between noisy and clean speech signals to improve speech. Ref. [22] design a DNN model to accurately predict the crop yield, and the result shows that the regression behavior of the DNN-based model in this scenario is better than shallow neural networks (SNNs). Ref. [23] conducted a comprehensive analysis of vanilla deep regression considering a large number of deep models with no significant difference in the network architecture and data pre-processing. Ref. [16] developed a nonlinear regression model with the technique of ResNet. By comparing to other nonlinear regression techniques, this work indicates the nonlinear regression model based on DNNs is stable and applicable in practice. Although DNNs show the progresses in regression with the deep hidden layers, it is limited that the feature of each layers is only used once and the feature reuse is not considered to improve the nonlinear approximation capacity [24].

Densely connected convolutional networks (DenseNet) introduce the concatenation shortcut into their frameworks [25]. The concatenation shortcuts play a significant role in realizing the feature reuse and the key information of initial input could be reserved and transmitted to the output, which makes DenseNet achieve good performance in applications [26,27,28]. DenseNet performs well in image processing because convolution is suitable for feature extraction of images with multiple channels. Usually, the input of images is high-dimensional and includes redundant information [29]. For instance, the input of a 96 × 96 pixels image with three channels would have 27,648 dimensions. If the number of neurons of the first hidden layer in the fully connected layer is the same as the input dimensions, the number of weights would be close to $10^{8}$, which is too enormous and significantly aggravates the computation efficiency. The convolution kernel is designed for reducing the repetitive parts among the variables and extract featured information. Hence, for image processing, convolution plays a significant role in reducing redundant information and improving the efficiency of the algorithm. However, ref. [16] found that convolutional neural networks may lose essential information from input features due to local convolution kernels and thus are not suitable for nonlinear regression. To tackle this issue, they introduce the so-called residual regression model by replacing convolution and pooling layers into fully connected (FC) layers in ResNet. By maintaining the shortcut within residual blocks, residual regression enhances data flow in the neural network and has been applied in many fields, such as computational fluid dynamics [30,31], computer-aided geometric design [32], and safety control in visual serving applications [33,34]. Besides the localness of convolution, the independence of input features also requires the replacement of convolution layers in a neural network regression model. To sum up, the motivations of replacing convolution layers into FC layers in this work are explained as follows:Convolution is a local operator. As introduced by [16] and presented in Figure 1, convolution is a local operator. The localness of convolution may result in the convolutional neural network losing essential information and even key input features from input variables. Therefore, the convolutional network is not good enough for nonlinear regression.Predictors of regression tasks are independent. However, convolution kernels are used to extract features from redundant information or correlated variables. Therefore, the convolutional neural network would lose key information from regression predictors. Specifically, the convolutional neural network often uses 2D convolution layers. The input one-dimensional feature vector needs to be reshaped into a matrix when the network takes a regression task. The entries in the reshaped matrix are seen as the corresponding gray values of pixels of a figure and all the input entries are independent. However, the neighboring gray values in a figure are often highly correlated. This dilemma requires the substitution of convolution layers into FC layers.

Inspired by the residual regression model, we propose a novel DenseNet model for nonlinear regression. Specifically, the new neural network retains the major architecture of DenseNet excluding convolution and pooling layers. Fully connected layers are the substitution of convolution and pooling layers in the dense block. Therefore, the conceptual architecture of our DenseNet regression model consists of a number of building blocks, and each building block is linked to the others by concatenation shortcuts. Through concatenation, the DenseNet regression model could realize feature reuse, and critical information could be reserved.

This paper is organized as follows. In Section 2, the architecture of DenseNet regression is clarified and in Section 3, we introduce the simulated dataset. In Section 4, we derive the results and have a discussion. Firstly, the performance of DenseNet with different depths is evaluated, and then we compare the results of optimal DenseNet regression model with other regression techniques. At the end of this section, we estimate the effect of input dimension on the performance of DenseNet regression. In Section 5, we use DenseNet regression to predict relative humidity. Finally, we conclude and propose the future work in Section 6.

## 2. The Architecture of DenseNet Regression

DenseNet introduces the concatenation shortcuts to enhance the feature reuse in each dense block, which is beneficial to reduce the possibility of losing critical information and increase the accuracy of DNNs.

Generally, based on the characteristics of DenseNet, the architecture of our proposed model employs concatenation and removes the convolutional part, so that DenseNet could better serve the nonlinear regression tasks. Figure 2 and Figure 3 demonstrate the details of architecture of this DenseNet regression model. Unlike convolutional DenseNet, the DenseNet regression model replaces convolution and pooling layers with fully connected layers in the dense block. Meanwhile, we maintain Batch Normalization layers from the original DenseNet to our novel networks. Batch Normalization is a typical regularization method with the advantage of accelerating the training process, reducing the impact of parameters scale, and allowing the utilization of higher learning rates [35].

The fundamental component of the DenseNet regression algorithm is the building block in Figure 2. There are three fully connected layers in this block, and each layer contains three operations, including batch normalization, dense, and ReLU activation function. Activation function is indispensable for the design of neural networks. Rectified linear unit (ReLU) activation function shown in Equation (Equation 1) is widely used in neural networks, as it does not activate all the neurons at the same time, immensely reducing the computation [36]. This is particularly beneficial to DenseNet regression where there are plenty of parameters to optimize. The ReLU function is shown as follows:(1)f(x)=max(0,x)

Another vital element in the building block is the concatenation shortcut, which is designed to append the input feature to the end of the output sequence in a building block. Furthermore, to satisfy the nonlinear regression tasks, a linear activation function is applied for the output (or top) layer. The number of layers in a building block is a hyperparameter and is determined by the accuracy and efficiency of neural networks. If there are few layers in the building block, the neural network would be too shallow and simple. Therefore, it is difficult to approximate nonlinear relationships. Nevertheless, if there are more layers in the building block, neural network parameters would be too large to optimize, which decreases the computation efficiency significantly. Meanwhile, under the same depth, neural networks with more layers in the building block have less concatenation shortcuts, decreasing the efficiency of feature reuse. Hence, the number of layers in the building block should be neither too small nor too large. Enlightened by the idea that there are three layers in each identity block and dense block in the optimal residual regression model [16], we make the input also go through three fully connected layers before concatenating in the building block.

Figure 3 shows an example of the architecture of a DenseNet regression model with four building blocks. The model begins with an input layer and is followed by four building blocks. Every building block is connected by concatenation shortcuts. The output layer is positioned in the end. Thus, the total layers of this model are 13. The feature reuse is reflected in the concatenation shortcuts or curve arrows in Figure 3. The features of initial inputs and outputs of each building block would be transmitted to the following output layer through concatenation shortcuts. For example, the input of building block 2 not only contains the output of building block 1, but also includes the initial input features. By analogy, the input of the top layer contains initial input features and the outputs of building block 1, 2, 3, and 4. Accordingly, the architecture of this model could keep feature reuse and enhance the performance of the neural network on nonlinear regression. It should be noted that the specific number of building blocks needs to be optimized under the given circumstances.

## 3. Simulated Data Generation

To better understand the performance of this novel regression algorithm before it is applied to specific fields, a simulated nonlinear dataset is introduced to test the algorithm. We set a maximum value of 1200 in the simulated dataset. Moreover, in order to enhance the degree of nonlinearity, we add two smaller values (400 and 800) in the dataset, and finally, we have a nonlinear piecewise function as shown in Equation (Equation 2). In this scenario, 10,000,000 samples are generated from Equation (Equation 2) where $x_{i}$ is uniformly distributed in the interval [0,4], i.e., ${\left\{x_{i}\right\}}_{i=0}^{6}∼\;U\left[0,\;4\right]$.
(2)$$y=\left\{\begin{array}{cc} \sum_{i=0}^{6}{\left(x_{i}\right)}^{i}, & \mathrm{if}\sum_{i=0}^{6}{\left(x_{i}\right)}^{i}<400;\\ 400, & \mathrm{if}\;400\leq \sum_{i=0}^{6}{\left(x_{i}\right)}^{i}<800;\\ 800, & \mathrm{if}\;800\leq \sum_{i=0}^{6}{\left(x_{i}\right)}^{i}<1200;\\ 1200, & \mathrm{if}\sum_{i=0}^{6}{\left(x_{i}\right)}^{i}\geq 1200. \end{array}\right.$$

Three hundred cases from the generated dataset are shown in Figure 4. In this work, 6,750,000 samples are employed for training the DenseNet nonlinear regression model, while 750,000 samples served as validation data. The remaining 2,500,000 samples are utilized for testing.

## 4. Results and Discussion

### 4.1. DenseNet Regression Model Specification

Before training, the original data generated by Equation (Equation 2) were standardized via the Min-Max scaler:(3)$${\hat{u}}_{k}=\frac{\max_{i}u_{i}-u_{k}}{\max_{i}u_{i}-\min_{i}u_{i}}$$
where $u_{k}$ is one of the sample data, ${\hat{u}}_{k}$ stands for standardized data of $u_{k}$, and *i* varies from one to the vector length.

Turning to the DenseNet regression model coding, Keras is employed as the application programming interface with TensorFlow as the backend. The program is run on Palmetto cluster of Clemson University. The computing environment is equipped with 10 CPUs and 10 GB RAM. The type of CPUs is Intel(R) Xeon(R) CPU E5-2640 V4 with 2.40 GH, and each core has two threads. Additionally, the Tesla P100-PCIE GPU is used for data training acceleration. It is produced by NVIDIA and has an 11.9 GB memory.

For the model optimization, mean squared error (MSE) is served as the loss function as shown in Equation (Equation 4). To minimize the loss function, the Adam method is applied in this work. The Adam method computes individual adaptive learning rates for different parameters and integrates the merits of AdaGrad and RMSProp method, which work well in sparse gradients and online and non-stationary settings, respectively [37,38,39]. The default learning rate of the Adam method is 0.001 in Keras, but in practice, we notice that the validation loss oscillates in training. Therefore, the learning rate is set to 0.0001 in this section, so that the neural network has a better convergence performance.
(4)$$\mathrm{Loss}=\frac{1}{N}\sum_{m=1}^{N}{\left(y_{m}-{\hat{y}}_{m}\right)}^{2}$$

Overfitting is a common issue when training machine learning models. It is probable that the loss gradually decreases during the training period while it rises in validation and testing. To prevent this and obtain a better model, early stopping strategy is used so that the neural network is very close to the epoch where the minimum validation loss arises. This strategy is effective and straightforward so that it is a popular regularization method in deep learning [40]. After adding this strategy to the neural networks, the algorithm would stop when no progress has been made over the best-recorded validation loss for some pre-specified number (or patience) of epochs.

Primary parameter settings for the numerical validation scenario in this paper are as follows. The input dimension is 7. The epoch number in training is 800, and the patience of early stopping is 100 epochs. The batch size for gradient descent is 5000. As mentioned above, the learning rate of the Adam method is 0.0001. The magnitude of training loss, validation loss, and testing loss are $10^{-4}$.

### 4.2. DenseNet Regression with the Optimal Depth

To evaluate the effect of depth and find its optimal value, multiple DenseNet regression models with different depths ranging from 4 to 37 were trained on the simulated dataset in this part. The corresponding training parameters and performance of different regression models are shown in Table 1. It was observed that with the rise of depth, the number of parameters and expected running time increased synchronously. When the depth was four, the DenseNet regression model had a high testing loss of $6.6064\times 10^{-4}$, which shows that the neural network regression model with four layers was too simple to address complex and nonlinear regression tasks. As the depth of the DenseNet regression model goes up, the testing loss goes down gradually. Remarkably, the testing loss reached the lowest point, $1.5194\times 10^{-4}$, when the depth was 19. Moreover, as the depth exceeded 19 and continued to increase, the testing loss went up again. For the model with a depth of 37, the training result showed OOM (stands for Out of Memory) due to the tremendous number of parameters and computation. The outcome indicates that the depth of DenseNet regression model should be neither too small nor too large. In conclusion, the DenseNet regression model with depth 19 had the minimum testing loss and the best performance on simulated data. Therefore, the depth of the optimal DenseNet regression model on the simulated dataset was 19. It is worth noticing that the testing loss had a slight difference as the depth ranged from 13 to 19. It is also noted that the data structures in real world applications may be different from the simulated dataset. Therefore, we recommend setting the value of depth in the range {13,14,⋯,19} if DenseNet regression is applied under real scenarios.

### 4.3. Comparisons with the Baseline

In this part, to evaluate the optimal DenseNet regression model, we also considered other regression techniques as the baseline on the dataset generated by Equation (Equation 2). The dataset was the same as Section 3. The 10,000,000 samples were generated by Equation (Equation 2) with 6,750,000 samples for training, 750,000 samples for validation and 2,500,000 samples for testing. Four linear models, including linear regression, ridge regression, lasso regression, and elastic regression, were applied to the simulated dataset. Nonlinear regression techniques incorporate conventional machine learning methods and artificial neural networks (ANNs), such as decision tree, support vector regression (SVR) machine, and deep residual regression. Neural networks contain the deep residual regression model and ANN Not Concatenated model. The residual regression model is a variant of ResNet. It replaces convolution and pooling layers with fully connected layers and has a good performance on nonlinear regression [16]. The ANN Not Concatenated model has the same structure as the optimal DenseNet regression model but has no concatenation shortcuts. This means that the depth of ANN not Concatenated model was 19. The epoch number and the patience of early stopping for neural networks were 800 and 100, respectively. The computing environment of residual regression model and ANN Not concatenated model is the same as DenseNet regression, modeling by Keras with TensorFlow as the backend. The four linear models, SVR model, and decision tree regression were built in Python with the scikit-learn package. The training results of different regression techniques are displayed in Table 2, including the comparison items, training time, validation loss, and testing loss. Table 3 lists the optimal hyperparameters of all regression models mentioned above. Grid search method is employed to optimize the value of hyperparameters of each regression model, excluding linear regression and the last three neural networks. Usually, it takes a long time to train a support vector regression model on a large dataset. Therefore, the SVR model was pre-trained before grid search. The pre-training gives smaller ranges of hyperparameters and thus improves the computing efficiency. Particularly, since the linear regression has no hyperparameters, the corresponding parameters were set as NA (NA stands for Not Applicable in Table 2 and Table 3).

The comparison results could be intuitively observed from the list of testing loss in Table 2. Generally, the first four linear models had more significant testing loss than nonlinear models, reaching the magnitude of $10^{-1}$, which turns out that these four linear techniques are not applicable for nonlinear regression tasks. Among the remaining nonlinear regression techniques, the testing loss of the support vector regression (SVR) machine also had a magnitude of $10^{-1}$. Although support vector regression has been pre-trained before grid search, it still had a significant testing loss, which shows that support vector regression is not suitable for large datasets. The artificial neural networks (ANNs) without dense concatenation shortcuts in Table 2 have the same depth as the optimal DenseNet regression model but has a higher testing loss compared to the optimal DenseNet regression. It was also observed that DenseNet regression presents the best behavior among all the regression models in Table 2, with the lowest testing loss ($1.5194\times 10^{-4}$). This shows that concatenation shortcuts could enable feature reuse and keep critical information, and thus have a critical effect on the performance of DenseNet regression. Furthermore, one could note that the residual regression model had the second-smallest testing loss ($2.5931\times 10^{-4}$) behind the DenseNet regression model. This is because the addition shortcuts in residual regression enable data flow and thus makes the model have a better performance. However, although residual regression can bypass addition shortcuts, the identity blocks and dense blocks are not densely connected. This illustrates that the outperformance of DenseNet regression to residual regression is due to the concatenation shortcuts and dense connection. Therefore, one could conclude that the topology of neural network and the method of connection (addition or concatenation) are essential to regression success. In conclusion, DenseNet regression is suitable for tackling nonlinear regression problems with high accuracy.

### 4.4. The Effect of Input Dimension

It is known that the number of parameters in a deep neural network is related to input dimensions. The number of parameters becomes more remarkable as the neural network gets more input variables, and thus it takes more time for the algorithm to optimize and obtain a good validation. It is also noted that the computation progress even has OOM (Out of Memory) errors when the magnitude of parameters reaches $10^{8}$ under the given computation environment and simulated dataset, as shown in the depth optimization part of Section 4.2. Therefore, to enhance the computational efficiency, the input dimension should be limited. Table 4 lists the input dimensions of the optimal DenseNet regression model and the corresponding number of parameters. The optimal depth of the used neural network here was 19. In Table 4, the magnitude of parameter number reached $10^{5}$ when the input dimension was 5. However, there was a sharp increase reaching 1,663,801 in the number of parameters as the input dimension extends to 20. Dramatically, when the input dimension is 50, the magnitude of parameter numbers becomes $10^{7}$. Nevertheless, when the input dimension varies from 50 to 100, the magnitude of parameter numbers does not change and is still kept at $10^{7}$.

In conclusion, as the input dimension increases, the number of parameters goes up. Especially when the input dimension exceeds 80, the number of parameters increases sharply. If the input dimension reaches 200, the magnitude of parameter number would be $10^{8}$. In contrast, according to Table 1, the outcome shows the out of memory (OOM) error if the magnitude of parameters reaches $10^{8}$. For the sake of computational efficiency, we conservatively suggest that if the computing capacity is as limited as our work, the input dimension for the optimal DenseNet regression model with depth 19 should be under 200 to avoid out of memory errors.

## 5. Application of DenseNet Regression on Climate Modeling

In recent years, machine learning has been successfully applied in atmospheric and environmental science, see [41,42,43,44,45,46,47,48,49] for more details. In this section, DenseNet regression is also employed on climate modeling. Similar to the application part of [16], DenseNet regression is used to approximate the nonlinear relationship between relative humidity (RH) and other meteorological variables. Relative humidity is defined as the ratio of the water vapor pressure to the saturated water vapor pressure at a given temperature. It has a critical effect on cloud microphysics and dynamics, and hence plays a vital role in environment and climate [50]. However, the relative humidity is not accurate under vapor supersaturation circumstances since the formation of cloud condensation nuclei needs water vapor to be supersaturated in the air, and there is no widely accepted and reliable method to measure the supersaturated vapor pressure accurately at present [51]. To address this issue, this section uses the DenseNet regression model with optimal depth to quantify the nonlinear relationship between relative humidity and other environmental factors.

Our data is from ERA5 hourly reanalysis datasets on the 1000 hPa pressure level of ECMWF (European Centre for Medium-Range Weather Forecasts) [52]. The input features of DenseNet regression are temperature (T) and specific humidity (q). The response variable is relative humidity. The dataset is selected from 00:00:00 a.m. to 23:00:00 p.m. on 1 September 2007 with a spatial resolution of $0.25^{\circ }\times 0.25^{\circ }$, and the spatial range of the dataset is global. There are 24,917,709 samples in total. The optimal DenseNet regression model with depth 19 is trained on 16,819,454 samples and validated on 1,868,828 samples. The remaining 6,229,427 samples are testing data. The batch size is 20,000, and the setting of other parameters are the same as the optimal DenseNet regression setting in Section 4. Furthermore, the consequence shows that the correlation coefficient (ρ) for the fitted values and observed values of testing data is 0.9095, which is shown in Figure 5. This indicates that the fitted values and observed ones are highly correlated. The average relative error in testing for relative humidity is 6.23%, verifying that the DenseNet regression model behaves excellently in practice.

To further evaluate the performance of DenseNet regression, a similar dataset on another day was tested on the above DenseNet RH regression model. The dataset was selected at 00:00:00 a.m. of the next day (2 September 2007) and was still on the 1000 hPa pressure level. The spatial range of the data was global and the resolution was $0.25^{\circ }\times 0.25^{\circ }$. There are 1,038,240 samples in total. The result is shown in Figure 6. Global distributions of temperature, specific humidity, and observed values of relative humidity are presented in Figure 6a,b,c, respectively. The predicted values of relative humidity are displayed in Figure 6d. Compared to the result on 1 September, the correlation coefficient (ρ) between observed values and predicted ones of RH on 2 Septmeber 2007 was 0.90, and the average relative error was 6.22%, which indicates that the performance of DenseNet regression model is stable in practice.

## 6. Conclusions and Future Work

The convolutional DenseNet behaves well in image processing. However, when it is applied to regression tasks, the independence of input features makes the convolutional neural network lose critical information from input variables. To address this issue, we develop a novel densely connected neural network for nonlinear regression. Specifically, we replace convolutional layers and pooling layers with fully connected layers, and reserve the DenseNet dense concatenation connections to enhance feature reuse in the regression model. The new regression model is numerically evaluated on simulated data, and the results recommend an optimal depth (19) and input dimensions (under 200) for the regression model. In addition, we compare DenseNet regression with other baseline techniques, such as support vector regression, decision tree regression, and deep residual regression. It turns out that the DenseNet regression model with optimal depth has the lowest testing loss. Finally, the optimal DenseNet regression is applied to predict relative humidity, and we obtain a high correlation coefficient and a low average relative error, which indicates that the DenseNet regression model is applicable in practice and could advance environmental data science.

In the future, we intend to apply the DenseNet regression model to the parameterization of the subgrid-scale process of large eddy simulation of turbulence at the atmospheric boundary layer. In addition, we will also employ the DenseNet regression to estimate global terrestrial carbon fluxes using net ecosystem exchange (NEE), gross primary production (GPP), and ecosystem respiration (RECO) from the FLUXNET2015 dataset. 

## Figures and Tables

**Figure 1 entropy-24-00876-f001:**
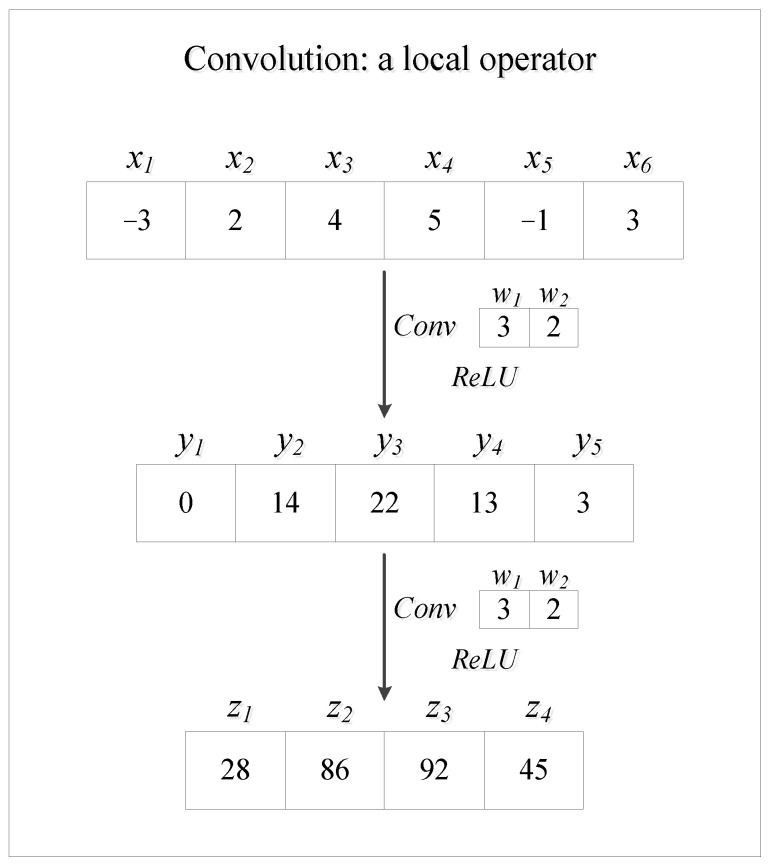
Convolution as a local operator. The input data are $\left(x_{1},\cdots ,x_{6}\right)$ and are convoluted with a convolution kernel $\left(w_{1},w_{2}\right)$. The stride of convolution is 1. Then, $y_{1}=\mathrm{ReLU}\left(w_{1}x_{1}+w_{2}x_{2}\right)=0$. When the vector $\left(y_{1},\cdots ,y_{6}\right)$ is convoluted again, we get $z_{1}=\mathrm{ReLU}\left(w_{1}y_{1}+w_{2}y_{2}\right)=\mathrm{ReLU}\left(w_{2}y_{2}\right)$. Here, when computing $z_{1}$, the neural network is losing information from $x_{1}$ since $y_{1}=0$. This illustrates that the localness of convolution may result in the neural network missing some input variables or essential information from input features. Thus, the convolutional neural network is not suitable for regression tasks.

**Figure 2 entropy-24-00876-f002:**
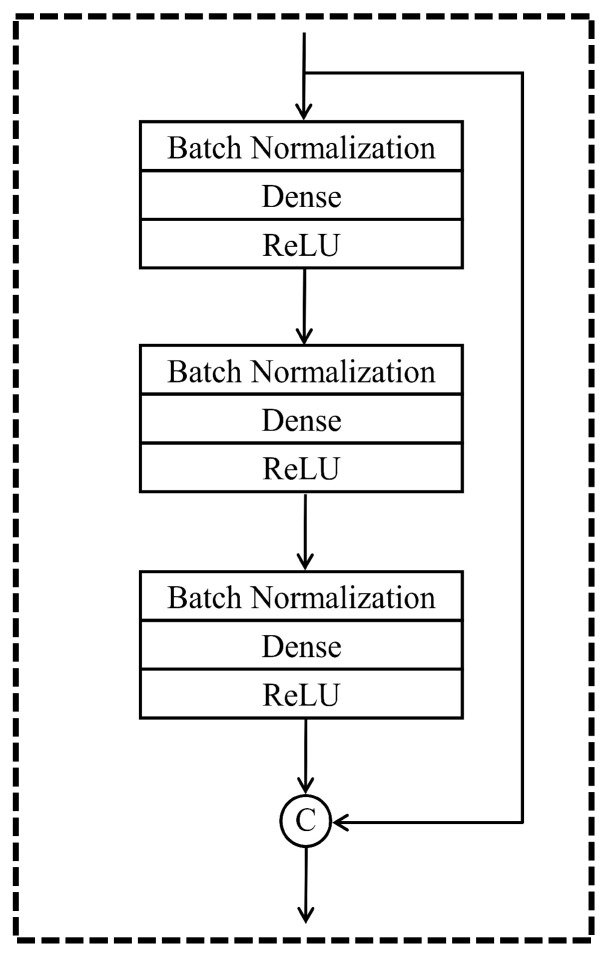
The building block of DenseNet for nonlinear regression. There are three hidden layers in each building block. At the end of the building block, there is a concatenation shortcut transmitting the information from top to bottom to guarantee feature reuse. ‘C’ in the diagram stands for concatenation.

**Figure 3 entropy-24-00876-f003:**
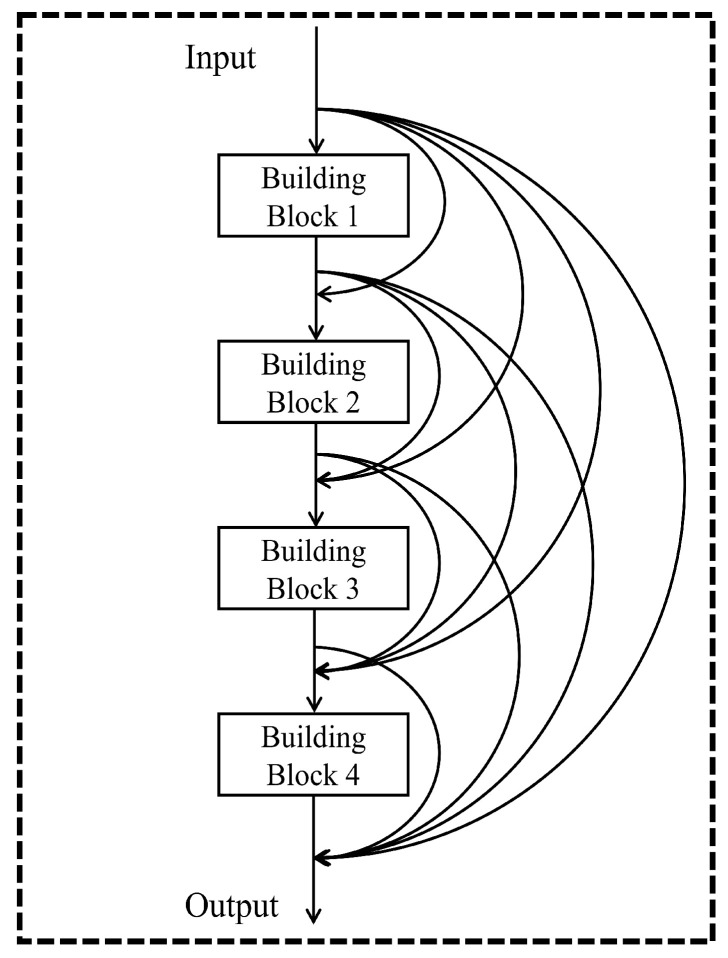
The architecture of a DenseNet regression model with 13 layers. The model has four building blocks. Each building block is linked to the others by a concatenation shortcut. The output layer is positioned in the end. Through concatenation, the features of initial inputs and outputs of each building block would be transmitted to the following output layer.

**Figure 4 entropy-24-00876-f004:**
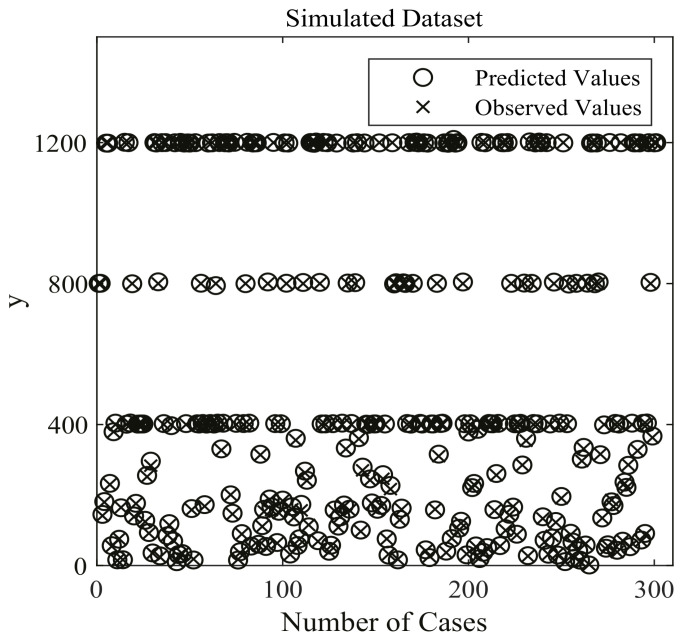
Three hundred cases of simulated Dataset for DenseNet nonlinear regression model. ◯ stands for the predicted values of optimal DenseNet regression model, and × stands for the true (observed) values of the simulated dataset.

**Figure 5 entropy-24-00876-f005:**
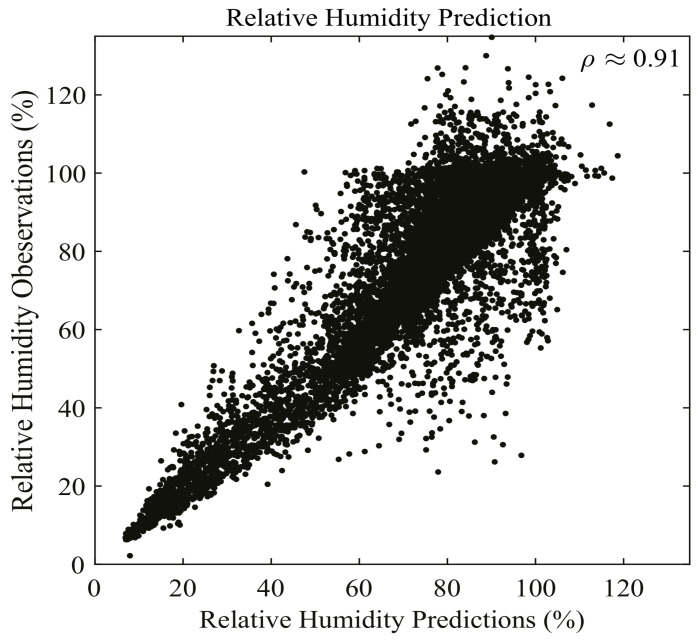
Performance of RH prediction on 1 September 2007 using DenseNet regression. The figure shows the first 20,000 cases of relative humidity prediction. The correlation coefficient ρ is approximately 0.91. This shows that DenseNet regression has high performance under real-world scenarios.

**Figure 6 entropy-24-00876-f006:**
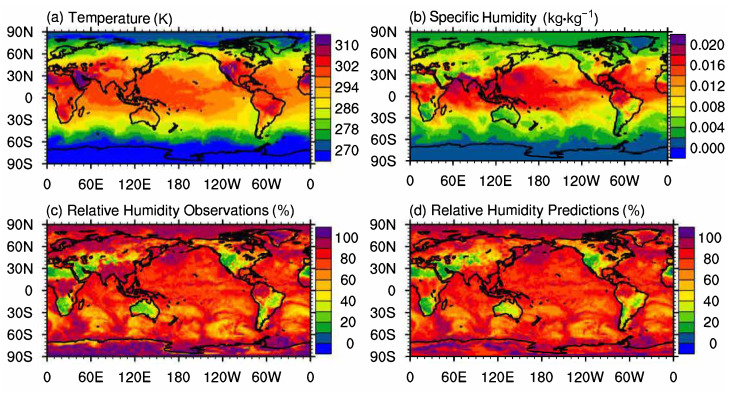
Performance of RH prediction on Sep. 2 using DenseNet regression. The figure shows the relative humidity prediction at 00:00:00 a.m. of the next day (2 September 2007) with the same DenseNet regression model in Figure 5. The dataset was still on the 1000 hPa pressure level. The spatial range of the data was global and the resolution was still $0.25^{\circ }\times 0.25^{\circ }$. The correlation coefficient ρ was 0.90, indicating that the performance of DenseNet regression is stable under real-world scenarios.

**Table 1 entropy-24-00876-t001:** Performance of DenseNet regression models with different depths.

Depth of DenseNet	Number of Parameters	Expected Time	Stopping Epoch	Training Time	Training Loss (10^−4^)	Validation Loss (10^−4^)	Testing Loss (10^−4^)
4	407	00:50:00	800	00:50:37	9.5808	6.5525	6.6064
7	1275	01:10:00	800	01:07:09	5.9951	3.5137	3.4935
10	4187	01:30:00	800	01:26:16	3.5369	2.4552	2.5214
13	14,715	01:50:00	800	01:49:37	2.9436	1.6492	1.6349
16	54,587	02:30:00	607	01:52:02	2.3569	1.8128	1.7606
**19 ***	**209,595**	**03:40:00**	**800**	**03:36:56**	**1.9152**	**1.4785**	**1.5194**
22	820,667	06:10:00	329	02:31:16	2.6473	2.0147	2.0633
25	3,247,035	11:50:00	404	05:58:19	2.2012	2.1022	2.1704
28	12,916,667	28:20:00	333	11:48:34	2.3818	3.4843	3.4365
37	822,656,955	OOM	OOM	OOM	OOM	OOM	OOM

* The optimal depth has a minimum testing loss. OOM stands for Out of Memory.

**Table 2 entropy-24-00876-t002:** Comparisons of DenseNet regression with the baseline models.

Regression Techniques Used	Stopping Epoch	Training Time	Training Loss (10^−4^)	Validation Loss (10^−4^)	Testing Loss (10^−4^)
Linear regression	NA	00:00:14	371.66	NA	371.52
Ridge regression	NA	00:00:28	371.65	NA	371.55
Lasso regression	NA	00:00:33	371.54	NA	371.87
Elastic regression	NA	00:29:57	371.57	NA	371.80
Support Vector Regression	NA	03:09:09	142.85	NA	143.01
Decision tree regression	NA	00:16:05	2.8724	NA	4.7552
ANN Not Concatenated	800	02:11:14	6.5255	4.0270	3.9924
Residual regression	687	02:39:22	2.3889	2.5674	2.5931
**DenseNet regression ***	**800**	**03:36:56**	**1.9152**	**1.4785**	**1.5194**

* DenseNet regression has a minimum testing loss. NA stands for Not Applicable.

**Table 3 entropy-24-00876-t003:** Optimal hyperparameters of regression techniques.

Regression Techniques Used	Name of Hyperparameters	Range of Hyperparameters	Optimal Hyperparameters
Linear regression	NA	NA	NA
Ridge regression	Penalty parameter of L^2^ norm α;	10^−10^, 10^−9^, 10^−8^ ⋯, 10^9^, 10^10^;	10^2^
Lasso regression	Penalty parameter of L^1^ norm α ;	10^−10^, 10^−9^, 10^−8^ ⋯, 10^9^, 10^10^;	10^−5^
Elastic regression	Penalty parameter α; Ratio of L^1^ norm ρ ;	10^−10^, 10^−9^, ⋯, 10^10^; 0.0, 0.1, ⋯, 0.9, 1.0;	10^−5^; 1.0;
Support Vector Regression	Penalty parameter α ; RBF kernel parameter γ ; Epsilon-tube parameter ϵ ; Maximum iteration *N*;	10, 10^2^, 10^3^; 10^0^, 10, 10^2^; 0.1; 3000;	10; 10; 0.1; 3000;
Decision tree regression	Maximum depth;	1, 2, ⋯, 13, 14;	14
ANN Not Concatenated	Depth	NA	19
Residual regression	Width; Depth;	NA	16; 28;
DenseNet regression	Depth	NA	19

**Table 4 entropy-24-00876-t004:** The effect of input shape.

Input Dimension	Number of Parameters
5	108,751
10	422,301
15	940,651
20	1,663,801
30	3,724,501
40	6,604,401
45	8,351,551
50	10,303,501
60	14,821,801
70	20,159,301
80	26,316,001
100	41,087,001
150	92,350,501
200	164,094,001

## Data Availability

Available online via https://cds.climate.copernicus.eu/cdsapp#!/dataset/reanalysis-era5-pressure-levels?tab=form (accessed on 21 May 2022).

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
