# Peer review of "Densely Connected Neural Networks for Nonlinear Regression"

_entropy, 2022, doi:10.3390/e24070876_

Round 1
Reviewer 1 Report
Overall, this is a very interesting paper. The authors point out the defects of the traditional CNN method in the field of nonlinear regression prediction, and put forward a regression model based on DenseNet. Then, the simulated data and the realistic meteorological data from ECMWF reanalysis are used to validate the method, which proves that the method can better provide a regression prediction for nonlinear datasets. The reviewers applauded the author's commitment to developing new techniques, as these studies would benefit the entire community. However, this paper is somehow weak in the validation part, and the reviewer suggests that the article be reviewed after a major revision.
Major comments:
The validation section is relatively weak, which is not conducive to readers to better understand the performance of this new and excellent method. The author uses the ECMWF data for testing, but does not provide information on the spatial range, spatial resolution and temporal resolution of the ECMWF data used. It is suggested to supplement these information. At the same time, the reviewer suggested adding two figures to better support the conclusions. One figure is suggested to show the flow of validation, which part of reanalysis data is used as the training set and which part is used as the test set. What is the predicted time step during the test? 1 hour or 6 hours? These information can be clearly represented by a picture. The second figure should give the predicted spatial distribution of relative humidity and the spatial distribution map obtained from original reanalysis data, which can more intuitively show the prediction effect, especially for scholars in the field of Meteorology without machine learning background.
Minor comments:
1. Lines 19-23: This is not a complete sentence, please reorganize it.
2. Section 2: why use 4 blocks instead of 3 or 5?
3. Table 1: how to understand the relationship between the depth in the Table and the number of blocks and layers in Figure 3?
4. Line 290: ECMwF → ECMWF
5. Reference: in recent years, there are many applications of machine learning in the field of atmospheric science. Considering that the test of this paper is also related to atmospheric science, it is suggested to add relevant references.
Reviewer 2 Report
The paper is a really good work showing a new regression methodology validated with respect to different regression methods. This validation was done with a proper database or a case study to curve fitting the relative humidity of moist air, which is a clear example of its application. The methodology was explained and developed in accordance with the scientific methodology to reach the objectives.The paper format is good, and the reference style is in accordance with journal indications. In general, it is a good research work well developed, explained and with an adequate paper format. In my opinion, it is adequate for publication as it is.
Author Response
Dear reviewer,
Thank you for your positive feedback about our manuscript. We have carefully proofread the manuscript again to minimize typographical and bibliographical errors. We deeply appreciate your patience and constructive comments.
Sincerely,
The authors

Round 2
Reviewer 1 Report
The authors have made a comprehensive response to the reviewer's comments, which has fully improved the quality of this manuscript. It is suggested to accept it in its current form.